# "Resilience amidst challenges": Healthcare users' experiences of access and utilisation of primary healthcare services during the COVID-19 pandemic in southwestern Uganda

Rachel Kawuma[1]*, Joseph Katongole[1], Abdmagidu Menya[1], Winnie Eoju[2],
Jonathan Kitonsa[3], Katherine Gallagher[4], Deborah Watson-Jones[5], Eugene Ruzagira[3,4]

**1** Social science department, MRC/UVRI and LSHTM Uganda Research Unit, Entebbe, Uganda,
**2** Uganda Virus Research Institute, Entebbe, Uganda, **3** Viral Pathogens Epidemiology and Interventions, MRC/UVRI and LSHTM Uganda Research Unit, Entebbe, Uganda, **4** Department of Infectious Disease, Epidemiology and International Health, Faculty of Epidemiology and Population Health, London School of Hygiene and Tropical Medicine, London, United Kingdom, **5** Department of Clinical Research, Faculty of Infectious and Tropical Diseases, London School of Hygiene and Tropical Medicine, London, United Kingdom

* rachel.kawuma@mrcuganda.org

## Abstract

Primary healthcare (PHC) systems are the first point of contact for healthcare users and are critical for disease prevention and early treatment. The COVID-19 pandemic overwhelmed PHC systems globally, including in sub-Saharan Africa, significantly impacting access to essential services. We describe the experiences of community PHC users' accessing and utilizing health services at 15 purposively selected health centers during the COVID-19 pandemic in Masaka district, southwestern Uganda. A qualitative study (13th-August-2021–4th-November-2021) was embedded with in a large investigation evaluating the impact of COVID-19 on PHC services across three African regions. We conducted 28 in-depth interviews (IDIs) and 2 focus group discussions (FGDs) among adult (females and males) healthcare users (aged ≥18 years). Data was analyzed by themes and charted against the Andersen's Behavioral Model for healthcare utilization. The analysis revealed that structural adjustments to enforce COVID-19 protocols, such as handwashing stations and reduced waiting times, were viewed positively. Facilitators of healthcare access included community sensitization, positive attitudes of health workers, and users' determination despite movement restrictions. However, barriers included fear of infection, medication shortages, reduced clinic hours, and the prioritization of COVID-19 care over other services. Participants adapted by obtaining travel permission letters and using herbal remedies when conventional care was inaccessible. Our key finding is that despite significant disruptions to PHC services during the COVID-19 pandemic, healthcare users demonstrated resilience in accessing care. However, systemic challenges, such as workforce shortages and limited essential medical supplies, highlighted

**Data availability statement:** All relevant data are within the paper and its Supporting Information files.

**Funding:** This project was funded by a UKRI (MRC), DHSC (NIHR) research grant (GEC1017, MR/V029363/1, PI Katherine Gallagher). The funder was not involved in study design; in the collection, analysis, and interpretation of data; in the writing of the report; and in the decision to submit the article for publication.

**Competing interests:** The authors have declared that no competing interests exist.

vulnerabilities within the PHC system. Addressing these structural barriers and investing in more resilient and adaptive healthcare systems is crucial for ensuring equitable access to essential health services during future public health crises.

## 1. Introduction

The Primary health care (PHC) system is often the first point of contact through which people interact with the health system and if well attended, PHC is cost effective as it mitigates expenses that would otherwise be incurred if disease is not detected early [1]. Therefore, a decrease in the provision of PHC services may have severe consequences, especially in low and middle income countries (LMICs) such as Uganda, where healthcare delivery systems are known to be inadequate [2]. Globally, countries relied heavily on the PHC system in their rapid response to the COVID-19 pandemic [3,4]. There is evidence that despite Sub-Saharan Africa (SSA) reporting lower COVID-19 cases and deaths compared to developed countries, the impact of the COVID-19 pandemic overwhelmed the PHC system in the region and affected delivery of essential services [5–7].

Uganda uses a decentralized healthcare system, with Village Health Teams (VHTs) serving as its foundational tier (Fig 1).

The healthcare system in Uganda is organized into different administrative levels, ranging from the village level to the national referral hospitals at country level [8]. PHC services are provided at public health facilities classified as Health Centers (HC) with the most basic healthcare at level II, then III, HC IV and the district referral hospital [9]. During the pandemic, all health centers were equipped with personal protective equipment (PPE) appropriate for COVID-19 and all health workers were prepared to identify and manage mild COVID-19 cases but to refer severe COVID-19 cases to referral health facilities, in this case at Masaka regional referral hospital (MRRH) [9]. Despite having a strategy to ensure continuity of health care service provision during the pandemic, Uganda experienced a decline in utilization of PHC services, including sexual and reproductive health (SRH) and maternal and child health (MCH) services [2,7,9,10]. This decline was largely attributed to stringent national lockdown measures, such as curfews, travel restrictions and border closures, which limited access to these services [9,11]. Additional factors, including fear of contracting COVID-19 at healthcare facilities, drug shortages, and re-allocation of healthcare staff to the COVID-19 response, further exacerbated the reduced use of PHC services [12].

While several studies have assessed the impact of the COVID-19 pandemic on routine PHC service delivery and utilization in resource-limited settings such as Uganda [13,14], most relied on secondary data from the national health information systems [2,4,6,15,16]. Other studies collected self-reported data on experiences of PHC services through online surveys [7,16,17]. The few studies that collected primary data mostly focused on specific populations, such as adolescents and young people [17] or pregnant women [18], or investigated the utilization of specialized services such as SRH and MCH services [19].

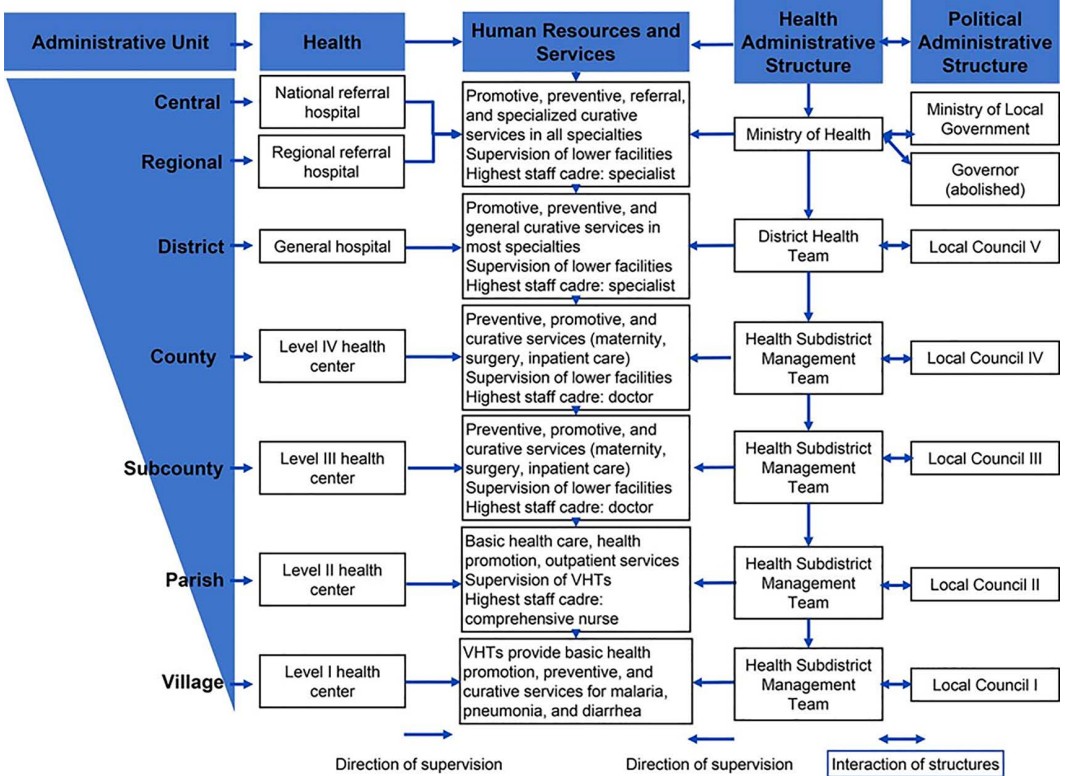

**Fig 1. Uganda healthcare system structure.**

This analysis is part of a qualitative sub-study within a larger study that evaluated the impact of COVID-19 on PHC provision and utilization in three distinct settings of Central Africa (The Democratic Republic of Congo), East Africa (Uganda) and West Africa (Sierra Leone) [10]. In this paper, we document the experiences of community PHC users' in accessing and utilizing healthcare services during the COVID-19 pandemic in Masaka district, southwestern Uganda.

### 1.1. Conceptual framework

We applied the Andersen's Behavioral Model (ABM) for healthcare utilization [20] to frame our study findings. This model has been widely used in previous studies to assess both individual and structural determinants of healthcare service utilization across various populations and settings [21–23]. The model comprises different constructs including the external environment (such as location of health facility, and the COVID-19 pandemic), population characteristics (including pre-disposing characteristics such as age, marital status and level of education, enabling factors (facilitators and barriers to health utilization) and the need factor which is defined as the subjectivity with which people view and define their general health and how that drives them to seek healthcare), All these constructs collectively influence health behaviors and outcomes (i.e., perceived and evaluated health status). (Fig 2).

## 2. Materials and methods

### 2.1. Study setting and population

The main COVID-19 study took place at selected rural and urban PHC facilities in Masaka district, Uganda between July 2021 and May 2022. Masaka district has a regional referral hospital and 25 government PHC facilities including 2 HC IVs,

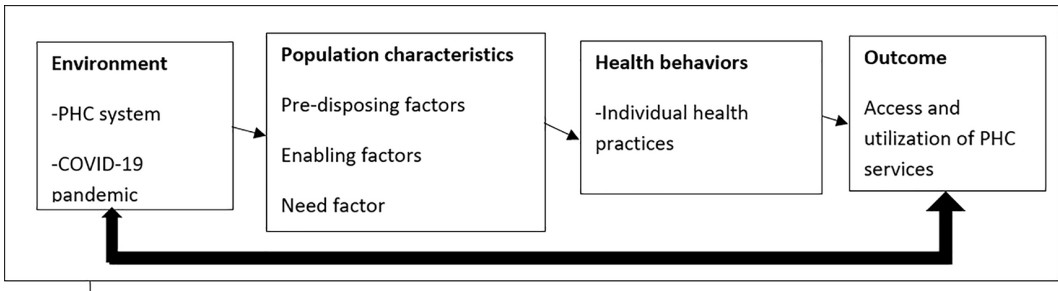

**Fig 2. An illustration of Andersen's behavioral model of health service utilization (ABM).**

9 HC IIIs and 14 HC IIs. All the 25 HCs were included in the larger study to evaluate the impact of COVID-19 on PHC provision and utilization [10]. The project aimed to examine how, and to what extent, the COVID-19 pandemic affected the provision and utilization of primary health care services. It had five objectives, including documenting community members' experiences in accessing primary health care during the outbreak and identifying the barriers and facilitators to its utilization. Another objective, which we hope to address in another manuscript, was to document the experiences of primary health care workers in delivering care during the outbreak, along with the barriers and facilitators they encountered. In this qualitative sub-study, we purposively recruited adult (≥18 years) male and female healthcare workers and healthcare users who were willing and able to provide informed consent. This paper focuses on findings from healthcare users only.

## 2.2. Sampling and recruitment

Multi-stage sampling was used. First, we systematically selected 15 HCs from the 25 government HCs at all the three tiers of HC IV, III and II. Fifteen out of 25 health facilities were selected to ensure variation in community contexts, facility levels, and geographical settings within Masaka District. Facilities with significant service disruptions were prioritized to understand how the pandemic affected care provision and access. The mix of facility levels (HC II, III, and IV) captured differences in capacity, infrastructure, and services. The final sample included 2 HC IVs, 8 HC IIIs, and 5 HC IIs, half in urban areas and half rural. The inclusion of both urban and rural health facilities explored how location influenced healthcare users' experiences and community health care access during the pandemic, notwithstanding how the different socio-economic and geographic realities affected access to care. Given logistical and safety constraints during the pandemic, the selected facilities were those that were logistically feasible to reach by the research team within the study period, while still maintaining health protocols and representation across the district. At each facility, the study team, in collaboration with facility health workers, obtained lists of individuals attending the facility on a given day when the qualitative study team visited to identify potential participants. Sampling criteria included age, gender, and the reason for seeking healthcare services. Identified individuals were then approached by a member of the study team, who provided them with detailed information about the study and requested their permission to take part in the study. Those who agreed provided written informed consent and took part in either an in-depth interview (IDI) or focus group discussion (FGD).

Each consenting healthcare user participated in a single IDI on the same day that they sought healthcare services at the facility. Additionally, two FGDs were conducted: one with eight male participants, and another with seven female participants, both at a HC II. The IDIs and FGDs explored community perspectives on access to and utilization of health services. Key areas explored included community members' experiences of COVID-19, their perceptions of the risk of COVID-19 and of seeking primary healthcare, their trust in healthcare services and how the pandemic influenced their attitudes and behavior towards PHC services.

## 2.3. Data collection

Qualitative data collection was conducted during the second wave of the COVID-19 pandemic in Uganda, between 13th-August-2021–4th-November-2021, a period marked by severe cases of COVID-19 and restrictive lockdown measures. To ensure safety, the study team adhered strictly to safety protocols, including wearing face masks, practicing hand hygiene, and maintaining social distance of at least 2 meters apart as per the COVID-19 SOPs, during face-to-face data interactions. All IDIs and FGDs were conducted by trained male and female research assistants using a topic guide (S1Text). The sessions were conducted in Luganda, the local language spoken by all participants.

All the interviews were audio recorded, except for those conducted at one health facility located in a correctional center, where written notes were taken and expanded shortly after the IDIs due to correctional facility protocols prohibiting the use of electronic devices. Written informed consent was obtained before conducting the IDIs and FGDs, which took place in private locations within the health facilities, often chosen by the participants. The IDIs typically lasted 45–60 minutes, while the FGDs ranged from 60 to 90 minutes. Brief notes were also taken during the sessions to capture non-verbal cues that could not be recorded. Regular de-briefing meetings were held between the interviewers and the study coordination team to ensure data completeness and identify areas for improvement.

## 2.4. Data management and analysis

To protect participants' identities and ensure confidentiality, interview transcripts were anonymized using unique study identifiers, and notebooks were securely kept under lock and key. Audio recordings and written interview notes were stored on password-protected computers and backed up on secure MRC/UVRI and LSHTM Uganda Research Unit servers, accessible only to the study team. The interviews were translated into English by the same research assistants who conducted them before the data coding process commenced.

Data analysis followed the principles of thematic content analysis [24], beginning during the data collection process. Emerging codes were identified and discussed during the debriefing meetings to ensure familiarity and refinement of themes. Coding was conducted using both deductive and inductive approaches. First, the study team developed a coding book based on the study objectives and used it to code a few transcripts. Three team members (JK, AM and WE) conducted the first level of manual analysis by carefully reading through two similar scripts and identified common codes. To increase inter-coder reliability and validity, regular de-brief meetings were conducted to discuss coding decisions on the same data segments, clarify understanding of the codes and refine their interpretations. During this process, additional codes emerging from the data were identified and those with similar or closer meaning were combined to form broader themes used to develop a coding framework (S 2 Text). Each theme was charted against the ABM constructs (Table 1). Nvivo 12 software was used to code the interviews, and each of the coders was assigned equal number of transcripts to code and thereafter, the three codebooks were merged into a master code book.

## 2.5. Ethics statement

Approval for the qualitative sub-study was obtained as part of the larger COVID-19 study [10]. This study was reviewed and approved by the Uganda Virus Research Institute Research Ethics Committee (Reference number: GC/127/821), the Uganda National Council for Science and Technology (Reference number: H1430ES) and the London School of Hygiene and Tropical Medicine Research Ethics Committee (Reference number: 22726–1). Participants were approached and invited to take part by the local research team and asked to provide written consent to participate in either IDIs or FGDs.

## 3. Results

### 3.1. Socio-demographic characteristics

Altogether, 43 healthcare users aged eighteen years and above participated in IDIs and FGDs and females were more than half of the population. Sixteen (37%) of these were aged between 25–34 years old. Most participants 21 (49%), had attained

**Table 1. Themes explaining healthcare users' experiences of access and utilization of primary healthcare services (PHC) during the COVID-19 pandemic in Masaka district, southwestern Uganda.**

| Themes | Sub-themes and codes | Anderson's Construct/s |
|---|---|---|
| Structural modifications to enforce COVID-19 SOPs | • Physical structural changes<br>• Operational modifications<br>• Enforcing the COVID-19 SOPs | External environment<br>Enabling factors |
| Facilitators to healthcare seeking | • Knowledge and sensitization about COVID-19<br>• Resilience amidst challenges<br>• Role of the health workers | Enabling factors<br>Need factor |
| Barriers to healthcare seeking | • Prioritising COVID-19 services at the expense of other essential services<br>• Fear of COVID-19 infection<br>• Lack of essential drugs<br>• Reduced clinic operating hours | Need factor |

primary education as their highest level of schooling, and 35 (81.4%), were married with children. Among those who took part in IDIs, a significant proportion 20 (64%), were engaged in farming, both for subsistence and commercial purposes, and most 24 (86%) had lived in these communities for longer than one year. The participants sought healthcare services for various reasons, including 11 who sought HIV care and treatment (mainly ART refills), 8 sought treatment for illnesses such as respiratory tract infections, general body weaknesses, and asthma, 6 for antenatal care and 3 came for infant immunization. (Table 2).

In the following section, we present our findings divided into three themes namely, structural and operational modifications to enforce COVID-19 SOPs, facilitators, and barriers to healthcare seeking and their sub-themes.

**3.1.1. Structural and operational modifications to enforce COVID-19 SOPs.** This theme explores the perceptions of participants regarding the structural and operational changes which were introduced at health facilities to contain the spread of COVID-19. These included re-arranging the physical spaces to encourage social distancing, introducing more hand washing facilities to encourage regular hand washing, ensuring wearing of face masks and reducing the times spent at the facility. Participants had varied perceptions of these changes as discussed in the following sub-sections.

**3.1.1.1. Physical structural changes:** The physical set up at some health facilities was modified by erecting tents to increase sitting space and minimize the risk of overcrowding, installing extra hand washing stations to promote hand hygiene and frequent cleaning of the facilities was emphasized. Participants noted this and it possibly contributed to their confidence in accessing healthcare during the pandemic.

*"Anyway, some health facilities were clean before the pandemic; however, this was not consistently done all the time and whenever facilities at the health centers got dirty, they would not be cleaned again until the end of the day. But hygiene has improved in health facilities and there is provision of water for washing hands."* (IDI-Female, 21 years, HC II)

However, concerns were raised regarding the new clinic setup which required patients to sit and be attended to in open spaces. This arrangement raised feelings that privacy was compromised.

*"Many patients do not like sitting in the open and wait to be attended to because they feel that their right to privacy is being infringed upon, especially when it concerned ART refills"*. (IDI-Female, 31 years, HC III).

In addition, participants also noted that the new setup diminished the intimate relationship healthcare users previously had with the healthcare workers, as they were no longer able to share private information in a confidential setting.

*"Apparently, you cannot even come closer to the health worker. All they tell you is to keep your distance and avoid coming closer to them. At times, you may have some private information to discuss with the health worker,*

**Table 2. Socio-demographic characteristics of healthcare users who took part in the qualitative sub-study in Masaka, Southwestern Uganda (August–October 2021).**

| Variable | IDIs = 28 (%) | FGD = 15 | Total = 43 (%) |
|---|---|---|---|
| **Sex** | | | |
| Male | 12 (43.0) | 8 (53.3) | 20 (46.5) |
| Female | 16 (57.0) | 7 (46.7) | 23 (53.5) |
| **Age (years)** | | | |
| 18-24 | 3 (10.7) | 3 (20.0) | 6 (14.0) |
| 25-34 | 12 (42.8) | 4 (26.7) | 16 (37.0) |
| 35-44 | 8 (28.6) | 1 (6.7) | 9 (21.0) |
| ≥45 | 5 (17.9) | 7 (46.7) | 12 (28.0) |
| **Marital Status** | | | |
| Single | 5 (17.9) | – | 5 (11.6) |
| Married | 20 (71.4) | 15 (100.0) | 35 (81.4) |
| Divorced/Separated | 3 (10.7) | – | 3 (7.0) |
| **Education** | | | |
| No formal education | 1 (3.6) | – | 1 (2.0) |
| Primary | 16 (57.1) | 5 (33.0) | 21 (49.0) |
| ≥Secondary | 11 (39.3) | 10 (67.0) | 12 (49.0) |
| **Occupation** | | | |
| Farming | 20 (71.4) | 7 (46.7) | 27 (62.8) |
| Other* | 8 (28.6) | 8 (53.3) | 16 (37.2) |
| **Period living in the community (years)** | | | |
| <1 | 4 (14.3) | – | – |
| 1-5 | 10 (35.7) | – | – |
| 6-10 | 6 (21.4) | – | – |
| >10 | 8 (28.6) | – | – |
| **Reason for seeking care** | | | |
| Antenatal care | 6 (21.4) | – | – |
| HIV care and treatment | 11 (39.3) | – | – |
| Infant immunization | 3 (10.7) | – | – |
| Other# | 8 (28.6) | – | – |

*Engaged in other work besides farming such as saloon work, working in prisons, etc.

#Respiratory tract infections, general body weaknesses, asthma.

*but you end up concealing it because of that kind of social distancing at the health facility.*" (IDI-Female, 31 years, HCIII)

**3.1.1.2. Operational modifications:** Health workers were instructed to attend to patients promptly, to minimize their time at the facilities, reduce overcrowding, and ensure patients and healthcare workers leave before curfew time. Participants noted this as a significant change and appreciated the reduced waiting time at health facilities compared to the pre-pandemic period, as it allowed them to spend as little time as possible at the health facilities and go home before the curfew set.

"*The delays at the health facilities have also reduced, now when we come to the health*

*facility like for immunization; the health workers serve you very fast so you can leave the facility.*" (IDI-Female, 21 years, HCII)

**3.1.1.3. Enforcing the COVID-19 SOPs:**   We asked participants about their views on how the COVID-19 SOPs were enforced at the health facilities. Some noted that the enforcement of these SOPs was sometimes done in an unfriendly manner and affected access to healthcare as detailed below.

*"One time I came for my routine antenatal care services without a mask. The health worker ordered me to go back and wear a mask before she handles me… I had to devise means of buying the facemask to be attended to."* (IDI-Female, 30 years, HC III).

*"The government has encouraged people to wear masks in public; you cannot access this facility without a mask. Otherwise, the health workers cannot attend to you if you are not wearing a mask; you are forced to buy it to be working on."* (IDI-Male, 32 years, HCIV)

Overall, participants noted that the modifications to enforce COVID-19 SOPs which were introduced at the health facilities was a positive strategy to address concerns around COVID-19 infections. However, the challenge was always with the enforcement of these SOPs.

**3.1.2. Facilitators to healthcare seeking.**  We sought to understand what supported participants to seek health care services during the pandemic, and how the pandemic influenced their attitudes and behavior towards PHC services. We noted that having appropriate knowledge and updates about the pandemic, the health facilities remaining open, and the availability of health workers motivated people to seek healthcare. In addition, there was resilience to seek health care amidst challenges with transportation and curfew set up, as explained in the sub-sections below.

**3.1.2.1. Knowledge and sensitization about COVID-19:**   Participants mentioned that they acquired knowledge and awareness about COVID-19 through sensitization from village health teams (VHTs) and local leaders, as well as through community announcements, radio broadcasts, and television. As a result, they learnt about the symptoms of COVID-19, and preventive measures as noted by a 25-year-old IDI-female participant at a HC II:

*"In the market where I was working, the market leaders have been sensitizing people about the pandemic. There are also VHTs in every village who sensitize people within their areas of operation."*

*"The community leaders have tried to communicate covid19 messages to the community members. Though community gatherings were restricted during the pandemic period, they communicated these messages using the village community radios. The local leadership has been reminding community members of the SOPs like handwashing, wearing facemasks and keeping social distance".* (Female participant-FGD-HCIII)

Relatedly, we noted that the sensitization efforts led to a change in the community members mindset, resulting in an increased willingness to seek healthcare during the pandemic. For instance, when participants or their family members experienced COVID-19 related symptoms such as cough or flu, they reported seeking healthcare immediately, a practice they seldom followed before the pandemic.

*"Based on what I hear on the radio; I do not let my child have flu for more than 2 days without bringing them to the health facility because you never know where COVID-19 may come from."* (IDI-Male, 52 years, HC IV)

**3.1.2.2. The Role of health workers:**   Participants were asked to assess whether the attitudes of health workers contributed to their efforts to seek health. Many mentioned that health workers had a positive attitude which motivated them to seek services:

*"Honestly, I am impressed with the services, the health workers have a good attitude and care about the patients; unlike other government health facilities, what I found here is different from what I expected to get… I spent less than 30 minutes to be worked on".* (IDI-Female-30 years-HCIII)

*"Health workers have tried to maintain proper hygiene and sanitation in the health*

*facilities. They have also tried as much as possible to avail themselves at the health facilities to attend to the patients"*- (FGD-Females-HCII)

**3.1.2.3. Resilience amidst challenges:** Despite the challenges posed by the pandemic, participants demonstrated resilience in seeking PHC services. The lockdown measures, including restrictions on public transport and travel across district borders, presented significant barriers. At the time, individuals needed permission letters from local leaders to move. However, participants were able to obtain these letters to access healthcare, as narrated by one participant who needed ART refills:

*"I had to first go to the chairman and the police to get a letter so that I could use a boda boda (motorcycle taxi) which was not the case before COVID and also paid the transport fare which increased from 2000/= [<$1], to 7,000/= [~$ 2]".* (IDI-Female, 49 years, HCIV)

In addition, the transport costs to the health facilities more than doubled but participants devised ways to get the money and seek healthcare.

*"Before COVID-19, they used to charge us 1500 shillings [<$1] from here to the facility but now they want 6000 or 7000 shillings [~$3] and if you find a police man or a police check point you have to get off the motorcycle and walk passed the check point and he rides ahead before boarding again after the road block."* (IDI-Male, 52 years, HCIV)

Some participants who could not afford transportation costs devised alternative ways to access care at nearby health facilities, as illustrated in the following quote:

*"During the first wave of the pandemic, I was in district XY and needed to come to district YZ for my HIV drugs… but hiring a boda-boda (motorcycle taxi) was so expensive so I approached a health worker in the nearby government facility, explained to them my situation, and they managed to provide me with a one month's refill though I used to get a refill of about three to four months."* (IDI-Female,27 years, HC III)

**3.1.3. Barriers to seeking healthcare.** However, despite making efforts to access healthcare at these facilities, several barriers existed such as fear of contracting COVID-19, having few health workers, reduced clinic time due to curfews and lack of essential drugs. This resulted in either receiving no drugs, being served late, or in extreme cases turned away from services especially when vaccination was prioritized over other services, as explained in the following sections.

**3.1.3.1. Prioritizing COVID-19 services at the expense of other essential services:** The study took place during the second wave of the COVID-19 pandemic in Uganda, a period during which the health system was focused on prevention and management of COVID-19. While all participants were able to receive some PHC services, COVID-19 related activities, including screening, testing, and vaccination were prioritized. This compromised quality and attention to detail given to other healthcare users. For instance, one female, who had come to get her newborn immunized noted that she left without getting the service because she attended the facility on a day that vaccination against COVID-19 was being conducted.

*"I came to the facility alone with my newborn child at around 9.00am (to receive childcare services). That day I found that people were receiving their COVID-19 vaccination and the health worker in charge of immunization was too engaged with COVID vaccination. I waited at the facility for so long and started feeling back pain due to sitting. Then*

*I left at around 2.00pm since I was too tired waiting (without receiving the service she had come for).*" (IDI-Female, 31 years, HC III)

In addition, participants receiving their anti-retroviral treatment (ART) who constituted majority of our sample noted a reduction in the supply of drugs compared to the pre-pandemic period saying that:

"*Before the pandemic, they would give me drugs for 6 months but now I only get drugs for 2 months or less*" (IDI-Male, 44 years, HC III).

**3.1.3.2. Fear of COVID-19 infection:** Another barrier to seeking healthcare services was the fear of contracting the virus at healthcare settings.

"*The biggest challenge to accessing primary healthcare services during the pandemic to both health workers and users has been fear of contracting the virus and because of this, health workers do not work on people without masks*". (IDI-Female-30 years-HCIII)

This was partly due to rumors circulating within the community that there were COVID-19 cases in the health facilities. This resulted in some individuals, like the participant below to delay their visits to healthcare facilities.

"*People scared us that there were many cases of COVID-19 at the facility, and I feared coming for child immunization because I was protecting myself and my child from getting COVID-19. That is the sole reason why I have not been seeking health care from the facility and just resumed the immunization schedule today*" (IDI-Female, 30 years, HCIII)

This fear was compounded by the concern that individuals with COVID-19-related symptoms could be sent to isolation centers.

"*During the lockdown, since flu is one of the signs of COVID-19, people who had flu stayed with it in their homes because they feared visiting health facilities for fear of being put in isolation centers at the health facilities*" (IDI-Female,25 years, HCII)

**3.1.3.3. Reduced clinic operating hours:** While participants generally observed that health workers were available in the facilities, in some facilities they often reported late for work, yet participants had to leave early to observe curfew times. Therefore, participants who came early but did not find the health workers decided to leave without getting the service as described below:

"*I came with my daughter when she was supposed to be immunized very early in the morning at around 7:00am and we waited until 11:00am when I decided to go back home. It was later at 12:00pm when I saw the health worker going to the facility because my home is near the road, but I could not go back and that is why you met me here today*". (IDI-Female, 36 years, HC II)

In addition, health facilities closed earlier than usual during the pandemic to ensure both healthcare users and health workers could return home before curfew hours. Consequently, patients who arrived later in the day risked not receiving care, requiring them to return on another day. The inconvenience often led to frustration, as expressed in the following account:

"*Previously most people would get healthcare services between 7am to 6pm, but now most people including the health workers stop working at 2pm because of the travel restrictions. This poses a big limitation as most people*

*are cut off from getting healthcare services and must push the visit to the following day*". (IDI-Male 25 years, HC III)

**3.1.3.4. Lack of essential drugs:** Participants noted the lack of essential medicines at health facilities, and instead they were given prescriptions to go and buy them. This left patients disappointed, especially because they did not have money to afford treatment in private facilities as expressed below:

*"We come to the health facility; at times you have borrowed transport to come to the facility or paid a lot of transport fare. You follow the line, get tested and when you get to the pharmacy you are told the drugs are not available. I am talking about malaria drugs not the ARVs and you are asked to buy. The fact that I came to the health facility should mean something to you, otherwise if I had the money, I should have just bought the drugs from a clinic and not bother coming to the health facility."* (IDI-Male, 36 years, HCIV)

*"There was a time I came here to the health facility when I was bleeding heavily because of the family planning method that I was using, I did not get the drugs, instead I was given the prescription note to go and look for the drug from out (clinic or pharmacy). I had to look for money and buy the drug, but it took a while for me to get the money.* (IDI-Female, 36-year, HCIII)

As a result, some participants who were unable to access services at government facilities and did not have money to pay at the private clinics, resorted to using herbal treatments to manage symptoms such as cough and flu.

*"I used herbal concoctions as advised by friends and other community members because I had no option due to the prevailing situation… if there is no money to pay at the clinic, the only available option is the local herbs"*. (IDI- Female, 27-year-old, HC III)

We noted that several barriers due to restrictions on transport, fears of covid 19 infections and prioritizing the response to COVID-19 over other services affected access to health care services during the pandemic. As a result, people devised other means such as using herbs to treat symptoms as they presented.

## 4. Discussion

We have described how community healthcare users made efforts to access PHC services amidst the challenges during the COVID-19 pandemic. Using Andersen's Behavioral Model of health service utilization, we explored how the environment, characterized by the COVID-19 pandemic, interacted with enabling factors to impact healthcare users' access to and utilization of PHC services at government health facilities in Masaka district. Further still, the structural modifications at the health facilities to enforce COVID-19 SOPs such as installation of hand washing, re-arranging the physical spaces to encourage social distancing and wearing face masks among others, contributed to increased patient satisfaction, including faster and more efficient delivery service, improved hygiene practices at health facilities, and increased vigilance among people seeking healthcare for any symptoms associated with COVID-19 as was observed in other settings [25]

A key facilitator for accessing and utilizing PHC services during the pandemic was the resilience demonstrated by healthcare users, a finding consistent with observations from other studies [26]. Wiig. S., et al (2021) define resilience as the 'capacity to identify and handle disruptions, large or small, and invoke mechanisms for systems to 'bounce back' and establish or re-establish a 'new normal' situation [27]. Despite the challenges posed by government lockdown measures such as movement restrictions, extended curfews, and limitations on public transportation [11,15,28] healthcare users in this study demonstrated remarkable resilience, persisting until they were able to access the healthcare services they needed. Factors contributing to this resilience, as noted in previous studies [26,29], include the re-assurance that

individuals would receive the services that they needed at government facilities and the high costs of similar services in private settings, which were unaffordable for many.

Participants' resilience was in part supported by sharing of COVID-19 related information by leaders in the community and through media which prompted people to seek health services, in instances when they had symptoms related to COVID-19. In Uganda, the government sensitized the community through regular broadcasts on media, increased mass testing and enforced observation of SOPs at health facilities and in the public sphere [30]. Having knowledge about disease has been found to contribute to good health seeking behaviours [31]. Previous studies have noted that effective communication during period of risk builds trust [32] and could contribute to uptake of healthcare services as was observed in this study.

However, while we noted resilience among this population, there were some reduction in the overall utilization of services during the pandemic [10]. This could be attributed to the barriers both at individual and structural levels that hindered access to health services in this study. For instance, as was noted in previous studies [12,33], government's prioritization of the COVID-19 response over other PHC services led to significant disruptions in the delivery of non-COVID-19-related healthcare services. This was exacerbated by the few health workers at the facilities who had to respond to COVID-19 related services at the expense of routine services such as child immunization [34]. In such circumstances, participants were left unserved which increased their frustration. To ensure continuity of access to all services in the event of future emergencies, governments could consider creating specific spaces for both routine PHC as well as the emergencies and dedicating health workers to take care of different health care users.

Other barriers were fear of infection among users, especially those seeking services from facilities that were also providing COVID-19-related care which has similarly been noted elsewhere [9,35]. In addition, was lack of essential drugs at the facilities as documented in previous studies [5,11], yet similar services are expensive in private facilities [36]. Addressing these challenges will be crucial to ensuring the effective and equitable delivery of PHC services during future pandemics.

Finally, this study also underscores the lengths to which individuals want to maintain their health during the pandemic, leveraging the resources available to them. For instance, when participants were unable to access public health facilities and could not afford care at private health facilities, some resorted to self-medication and/ or use of herbal remedies, to manage common health issues such as cough and flu. Similar findings have been documented in other studies [37]. Self-medication can serve as a safe and cost-effective form of self-care [38]. However, it is important to do it responsibly to ensure optimal health outcomes while minimizing potential risks [39]. On the other hand, while they are known to treat some ailments like common viral infections [40], there are conditions, such as new diagnoses of diseases which herbal remedies will not treat. Therefore, the fact that some participants resorted to using herbals during the pandemic represents some unmet need and possible delay in effective diagnosis and treatment. This highlights the need for governments to establish robust systems to monitor and understand citizens' health-seeking behaviors, which can inform the design of inclusive and effective healthcare interventions.

### 4.1. Strength and limitations

A key strength of the study was its adaptation of Andersen's Behavioral Model of health service utilization, which provided a framework to assess the relationship between the environment (characterized by the COVID-19 pandemic) and enabling factors, including facilitators and barriers to accessing and utilizing PHC services, using this framework enhances the rigor and depth of interpretation. However, our adaptation was limited to a few constructs of the model, as we did not measure health outcomes in this study. Consequently, we did not apply the model in its entirety to our specific setting. In addition, by sampling participants from those seeking health care at the time of conducting the study, we were able to capture experiences of people with varied characteristics and experiences, thus giving us an opportunity to assess access and utilization of several health services such as maternal and child health care, HIV/AIDS and SRH compared to other studies

which assessed only utilization of one component of health care. Also importantly, by recruiting healthcare users on days when they came to the health facilities, we were able to observe their experiences in 'real time' and this enriched our data and offered insights into how public health interventions are received and understood. Finally, using both IDIs and FGDs provided an opportunity for deeper probing and facilitated the collection of rich data.

However, our inability to include individuals from the general community who were not accessing PHC services at the time limited the scope of insights regarding the wider community's experiences during the pandemic. We may not have captured the full extent of barriers to health seeking given that we were sampling from a population who had eventually successfully accessed the health center. on the other hand, by interviewing participants at the health facility, there was a missed opportunity to interrogate the impact of their natural setting including, family and the wider community, which could have influenced individual experiences of the pandemic. Furthermore, the scope of our data, collected only over three months (August to October 2021), may not capture evolving perceptions as the pandemic progressed. And generally, we cannot rule out social desirability in a study such as this when people are bound to respond in a manner that they perceive is the correct particularly regarding sensitive topics like trust in authorities. Potential for social desirability bias in participant responses may have stemmed from fear of losing access to health services when discussing challenges faced at the health facilities. Particularly because interviews were conducted at the health facilities where health workers could see them. Additionally, incarcerated participants may have feared retribution from prison authorities. To address this, we included probing and consistency check questions to help verify the reliability of participants responses. Finally, the findings are specific to Masaka district and may not fully represent other regions or broader populations which may limit the generalizable of our findings to the wider healthcare user population in Uganda and elsewhere. We tried to mitigate this by sampling our participants across different tiers of the health systems to pick varied perspectives.

## 5. Conclusion

This study highlights the influence of community perceptions on public health interventions. Specifically, our findings reveal that perceptions of health and well-being initiatives are shaped by trust in authorities, the clarity of public health messaging, and the socio-economic realities faced by communities. Participants expressed both support for and concerns about intervention measures, highlighting the need for transparency, cultural sensitivity, and sustained community engagement.

To build public confidence, addressing misinformation and government working closely with community leaders emerged as an important strategy to improve reception and effectiveness of interventions. Furthermore, the study emphasizes that future public health initiatives should prioritize two-way communication and actively involve communities in the planning and decision-making processes to enhance trust and compliance.

Lessons from this study could inform the development of more responsive, equitable, and community-centered public health strategies. By recognizing and valuing community voices, policymakers can better tailor interventions to meet diverse needs and improve health outcomes on a broader scale.

## Supporting information

**S1 Text. Qualitative interview/discussion topic guide.**
(DOCX)

**S2 Text. Code book.**
(DOC)

## Acknowledgments

The authors would like to thank all the participants who took part in this study for their time and sharing their experiences with us.

## Author contributions

**Conceptualization:** Rachel Kawuma.

**Data curation:** Rachel Kawuma, Joseph Katongole, Abdmagidu Menya.

**Formal analysis:** Rachel Kawuma, Joseph Katongole, Abdmagidu Menya, Winnie Eoju.

**Funding acquisition:** Katherine Gallagher, Deborah Watson-Jones, Eugene Ruzagira.

**Methodology:** Joseph Katongole, Abdmagidu Menya, Winnie Eoju, Jonathan Kitonsa, Katherine Gallagher, Eugene Ruzagira.

**Project administration:** Eugene Ruzagira.

**Resources:** Katherine Gallagher, Deborah Watson-Jones, Eugene Ruzagira.

**Supervision:** Jonathan Kitonsa, Eugene Ruzagira.

**Writing – original draft:** Rachel Kawuma, Joseph Katongole, Abdmagidu Menya, Winnie Eoju, Jonathan Kitonsa, Katherine Gallagher, Eugene Ruzagira.

**Writing – review & editing:** Rachel Kawuma, Joseph Katongole, Abdmagidu Menya, Winnie Eoju, Jonathan Kitonsa, Katherine Gallagher, Deborah Watson-Jones, Eugene Ruzagira.

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
