## [Decision Letter · Decision Letter 0]

11 Jul 2025

PGPH-D-25-01125

“Resilience amidst challenges”; Healthcare users’ experiences of access and utilisation of primary healthcare services during the COVID-19 pandemic in southwestern Uganda

Dear Dr. Kawuma,

Thank you for submitting your manuscript to PLOS Global Public Health. After careful consideration, we feel that it has merit but does not fully meet PLOS Global Public Health’s publication criteria as it currently stands. Therefore, we invite you to submit a revised version of the manuscript that addresses the points raised during the review process.

Kindly address the comments that the two reviewers have raised to improve the submission.

We look forward to receiving your revised manuscript.

Kind regards,

Ferdinand C Mukumbang, PhD

Academic Editor

Journal Requirements:

2. In the online submission form, you indicated that [The data supporting the conclusions of this article may be made available on reasonable request through the corresponding author].

a. In a public repository,

b. Within the manuscript itself, or

c. Uploaded as supplementary information.

Additional Editor Comments (if provided):

Reviewers' comments:

Reviewer's Responses to Questions

**Comments to the Author**

1. Does this manuscript meet PLOS Global Public Health’s publication criteria ? Is the manuscript technically sound, and do the data support the conclusions? The manuscript must describe methodologically and ethically rigorous research with conclusions that are appropriately drawn based on the data presented.

Reviewer #1: Yes

Reviewer #2: Yes

2. Has the statistical analysis been performed appropriately and rigorously?

Reviewer #1: Yes

Reviewer #2: Yes

3. Have the authors made all data underlying the findings in their manuscript fully available (please refer to the Data Availability Statement at the start of the manuscript PDF file)?

Reviewer #1: Yes

Reviewer #2: Yes

4. Is the manuscript presented in an intelligible fashion and written in standard English?

Reviewer #1: Yes

Reviewer #2: Yes

5. Review Comments to the Author

Reviewer #1: Peer Review Comments

1. Overall Impression

This is a well-written and important study that explores the experiences of healthcare users in accessing and utilizing primary healthcare services in southwestern Uganda during the COVID-19 pandemic. The study addresses a critical gap in the literature by providing qualitative insights into the challenges and resilience of communities in a resource-limited setting. The use of Andersen's Behavioural Model is appropriate and provides a useful framework for analysis. The findings are relevant, and the discussion is well-supported by the data. The study has clear policy implications and contributes to our understanding of how to strengthen healthcare systems in the face of future public health crises.

2. Strengths

• Relevance: The topic is highly relevant, given the ongoing impact of the COVID-19 pandemic on healthcare systems globally, particularly in LMICs.

• Qualitative Approach: The qualitative methodology is well-suited to exploring the complex experiences and perspectives of healthcare users.

• Theoretical Framework: The use of Andersen's Behavioural Model provides a strong theoretical foundation for the study and allows for a structured analysis of the data.

• Rich Data: The study collected rich, qualitative data through in-depth interviews and focus group discussions, providing detailed insights into the experiences of the participants.

• Clear Presentation of Findings: The results are presented in a clear and organized manner, with illustrative quotes that support the themes identified.

• Ethical Considerations: The study clearly outlines the ethical approvals and informed consent procedures.

• Policy Implications: The study highlights important policy implications for strengthening healthcare systems and ensuring equitable access to essential health services during future public health crises.

3. Areas for Improvement

• Sampling Strategy: While the purposive sampling strategy is appropriate for qualitative research, the authors should provide a more detailed justification for the selection of the 15 health centers. Were there specific criteria used beyond the three tiers (HC IV, III, II) and urban/rural location? Providing more context about the health centers would strengthen the methodology.

• Data Analysis: While the authors mention using thematic content analysis, providing more detail about the specific steps involved in the coding process would enhance the rigor of the analysis. For example, mentioning inter-coder reliability checks or how disagreements in coding were resolved would be beneficial.

• Limitations: The limitations section is well-written, but the authors could expand on the potential impact of social desirability bias. How might the participants' responses have been influenced by their desire to present themselves in a positive light or to please the researchers?

• Clarity on Healthcare Workers: The abstract mentions recruiting healthcare workers, but the paper focuses on healthcare users. Clarify why the healthcare workers were not included in the analysis.

• Table 2: The table is not well formatted and is difficult to read. Consider reformatting the table to improve readability.

4. Policy Implications

The study has several important policy implications:

• Strengthening PHC Systems: The findings highlight the need to invest in more resilient and adaptive primary healthcare systems that can withstand future public health crises. This includes addressing structural barriers such as workforce shortages and limited essential medical supplies.

• Community Engagement: The study emphasizes the importance of community engagement and sensitization in promoting access to healthcare services. Policymakers should prioritize two-way communication and actively involve communities in the planning and decision-making processes.

• Addressing Fear and Misinformation: The findings highlight the role of fear and misinformation in influencing healthcare-seeking behaviour. Policymakers should develop strategies to address these issues through clear and transparent communication.

• Ensuring Equitable Access: The study underscores the need to ensure equitable access to essential health services, particularly for vulnerable populations. This includes addressing barriers such as transportation costs and a lack of essential drugs.

• Monitoring Health-Seeking Behaviours: The study highlights the need for governments to establish robust systems to monitor and understand citizens' health-seeking behaviours, which can inform the design of inclusive and effective healthcare interventions.

5. Overall Recommendations for Revision

I recommend that the authors revise the manuscript to address the following points:

• Provide a more detailed justification for the selection of the 15 health centres.

• Expand on the description of the data analysis process, including details about the coding process and inter-coder reliability.

• Elaborate on the potential impact of social desirability bias.

• Clarify why healthcare workers were not included in the analysis.

• Reformat Table 2 to improve readability.

Overall, this study makes a valuable contribution to the literature, offering a significant insight. With the suggested revisions, the manuscript will be even stronger and more impactful. I recommend that it be accepted for publication after these revisions are made.

Reviewer #2: This study addresses an important topic regarding primary healthcare (PHC) access and utilization during COVID-19. However, the main message and current relevance are unclear, given that 4-5 years have passed since the pandemic ended.

The key question is whether these findings have implications beyond the pandemic context. The study's outcomes lack clear relevance to current healthcare conditions, and while we hope to avoid future pandemics, the results could be better framed to provide actionable recommendations for future preparedness and resilience in healthcare systems.

The findings would be more valuable if they explicitly addressed how lessons learned during COVID-19 can inform routine healthcare delivery and emergency preparedness strategies moving forward.

6. PLOS authors have the option to publish the peer review history of their article (what does this mean? ). If published, this will include your full peer review and any attached files.

**Do you want your identity to be public for this peer review?** For information about this choice, including consent withdrawal, please see our Privacy Policy .

Reviewer #1: **Yes: ** Awah Kum Tchouaffi

Reviewer #2: **Yes: ** Derbew Fikadu Berhe

---

## [Editor Report · Decision Letter 1]

29 Jul 2025

“Resilience amidst challenges”; Healthcare users’ experiences of access and utilisation of primary healthcare services during the COVID-19 pandemic in southwestern Uganda

PGPH-D-25-01125R1

Dear Ms Kawuma,

We are pleased to inform you that your manuscript '“Resilience amidst challenges”; Healthcare users’ experiences of access and utilisation of primary healthcare services during the COVID-19 pandemic in southwestern Uganda' has been provisionally accepted for publication in PLOS Global Public Health.

Best regards,

Ferdinand C Mukumbang, PhD

Academic Editor